# FGF1 Protects MCF-7 Cells against Taltobulin through Both the MEKs/ERKs and PI3K/AKT Signaling Pathway

**DOI:** 10.3390/biomedicines11071856

**Published:** 2023-06-29

**Authors:** Jakub Szymczyk, Aleksandra Czyrek, Jacek Otlewski, Malgorzata Zakrzewska

**Affiliations:** 1Department of Protein Engineering, Faculty of Biotechnology, University of Wroclaw, Joliot-Curie 14a, 50-383 Wroclaw, Poland; jakub.szymczyk@uwr.edu.pl (J.S.); jacek.otlewski@uwr.edu.pl (J.O.); 2Department of Protein Biotechnology, Faculty of Biotechnology, University of Wroclaw, Joliot-Curie 14a, 50-383 Wroclaw, Poland; aleksandra.czyrek@uwr.edu.pl

**Keywords:** breast cancer, FGF1, EGF, drug resistance, taltobulin, ERKs, AKT, MCF-7

## Abstract

Breast cancer is a widespread and complex disease characterized by abnormal signaling pathways that promote tumor growth and progression. Despite significant medical advances and the development of increasingly effective therapies for breast cancer, drug resistance and reduced sensitivity to prior therapies remain persistent challenges. Dysregulation of growth factors such as FGFs and EGF and their receptors is a contributing factor to reduced response to treatment, promoting cell survival and proliferation, metastasis, EMT or increased expression of ABC transporters. Our study demonstrates a protective role for FGF1 in MCF-7 breast cancer cells against taltobulin-induced cytotoxicity, mediated by activation of its receptors and compares its activity to EGF, another growth factor involved in breast cancer development and progression. The mechanisms of action of these two proteins are different: FGF1 exerts its effects through the activation of both ERKs and AKT, whereas EGF acts only through ERKs. FGF1 action in the presence of the drug promotes cell viability, reduces apoptosis and increases cell migration. Although EGF and its receptors have received more attention in breast cancer research to date, our findings highlight the key role played by FGFs and their receptors in promoting drug resistance to tubulin polymerization inhibitors in FGFR-positive tumors.

## 1. Introduction

Breast cancer, despite the development of a number of diagnostic methods for its early detection, is the second most lethal malignancy after lung cancer [1]. Its treatment involves a diverse combination of available approaches, including surgery, radiotherapy and pharmacological therapies such as hormonal therapy, immunotherapy and chemotherapy [1,2,3,4]. Deciding on a care option is a complex process that depends on a number of factors: tumor stage and subtype, potential side effects of therapies, patient characteristics and general health. Despite significant advances in medicine and the development of new and increasingly effective therapies, patients are still at risk of metastasis and the recurrence of their disease, which often shows resistance to previously effective forms of treatment [5].

Breast cancer, like other types of solid tumors, is a complex disease involving dysregulation of signaling pathways that promote tumor growth and progression [6]. Overactivation of growth factors such as fibroblast growth factors (FGFs) and epidermal growth factor (EGF) or their receptors, as well as amplification of signaling as a result of mutations in specific signaling proteins, are thought to be involved in its pathogenesis [7,8]. The PI3K/AKT/mTOR and MAPKs cascades, which play a critical role in regulating cell growth, proliferation and survival, have been identified as key in breast cancer development and progression [6]. In addition, many studies have suggested that FGF and EGF signaling are also involved in the development of drug resistance [5,9]. Mechanisms of resistance encompass activation of alternative signaling pathways, downregulation of expression of therapeutic targets, evasion of programmed cell death and upregulation of ABC transporters. A deep understanding of the interplay between growth factors and drug resistance mechanisms in breast cancer is essential to develop effective treatments and minimize the risk of disease recurrence.

In this study, we analyze how FGF1, compared to EGF, affects the survival of the MCF-7 model cell line, extensively applied in breast cancer research [10] and treated with taltobulin, a cytotoxin belonging to the widely used group of drugs that inhibit tubulin polymerization [11]. To verify the effects of both growth factors on drug-treated cells, we assessed cell viability, apoptosis progression and cell migration. We observed a stronger protective effect against drug-treated cells for FGF1 than for EGF. To elucidate the mechanisms of the differential protective effect of the two growth factors, we used inhibitors of the major signaling pathways activated by these proteins. Our results indicate differences in the activation of signaling pathways leading to the protective action of FGF1 and EGF. In the case of FGF1, both ERKs and AKT are involved, whereas only ERKs are involved in EGF action. The findings emphasize the significant involvement of FGF1 and FGFRs in drug resistance in breast cancer; however, the mechanisms underlying this phenomenon appear to be intricate and require further understanding. Nonetheless, comprehending these mechanisms could potentially pave the way for the development of more efficacious therapies for breast cancer in the future.

## 2. Materials and Methods

### 2.1. The Reagents

Primary antibodies: anti-phospho-AKT (Ser473) (p-AKT (S473)) (#9271); anti-phospho-AKT (Thr308) (p-AKT (T308)) (#9275); anti-AKT (AKT) (#9272); anti-phospho-p44/42 (Thr202/Tyr204) MAP kinase (p-ERK1/2) (#9101); anti-p44/42 MAP kinase (ERK1/2) (#9102); anti-phospho-p70S6 kinase (Thr389) (p-p70S6K) (#9205); and anti-poly-[ADP-ribose] polymerase (PARP) (#9542) were from Cell Signaling Technology (Danvers, MA, USA); anti-γ-tubulin (tubulin) (#T6557) was from Sigma-Aldrich (St. Louis, MO, USA). Anti-cancer agents and inhibitors taltobulin (HTI-286) were from MedChem Express (Monmouth Junction, NJ, USA); vincristine was from Selleckchem (Houston, TX, USA); paclitaxel, PD173074, gefitinib, LY294002 and UO126 were from Sigma-Aldrich. Horseradish peroxidase-conjugated secondary antibodies were from Jackson ImmunoResearch Laboratories (Cambridge, UK). Heparin was from Sigma-Aldrich.

### 2.2. Recombinant Proteins

Recombinant proteins: FGF1 was produced in-house as previously described [12]; EGF protein was obtained from M.C. Biotec Inc. (Nanjing, China).

### 2.3. Cell Lines

The MCF-7 breast cancer cell line was obtained from the American Type Culture Collection (ATCC, Manassas, VA, USA), and maintained in DMEM (Biowest, Nuaille, France), with 10% fetal bovine serum (Thermo Fisher Scientific, Waltham, MA, USA) and antibiotics (100 U/mL penicillin, 100 μg/mL streptomycin). Cells were cultured at 37 °C in a 5% CO_2_ atmosphere.

### 2.4. Cell Cytotoxicity Assay

MCF-7 cells were seeded in 96-well plates at a density of 1 × 10^4^ cells/well in DMEM supplemented with 10% FBS and antibiotics. After 24 h, cells were treated with 5 nM taltobulin (TLT), 20 nM paclitaxel (PTX) or 10 nM vincristine (VCR) in the presence or absence of 10 ng/mL of FGF1 or EGF and 10 U/mL heparin. When chemical inhibitors (100 nM PD173074 or 10 µM gefitinib) were used, they were added to the cells 15 min before the administration of the specified drugs and growth factors. After 48 h incubation, alamarBlue Cell Viability Reagent (Thermo Fisher Scientific) was administered according to the manufacturer’s instructions. The fluorescent reduced form of the dye was measured at 590 nm after excitation at 560 nm using an Infinite M1000 PRO plate reader (Tecan, Männedorf, Switzerland). The cytotoxic effect of the drugs was normalized to untreated cells. All assays were repeated at least three times (n = 3) in triplicate.

### 2.5. Cell Migration Assay

Cell migration was analyzed using the IncuCyte^®^ Cell Migration and Invasion System (Essen BioScience, Royston, UK). MCF-7 cells (at a density of 4.5 × 10^4^ cells/well; in DMEM with 10% FBS and antibiotics) were seeded on a poly-D-Lysine coated 96-well IncuCyte^®^ ImageLock plate and scratched with the IncuCyte^®^ WoundMaker. Cells were then treated with 5 nM TLT and stimulated with 10 ng/mL of FGF1 or EGF and 10 U/mL heparin in the presence or absence of a specific inhibitor (100 nM PD173074 or 10 µM gefitinib). Wound images were automatically acquired every 2 h. Data were analyzed with the IncuCyte^®^ ZOOM GUI Version: 2018A Software package. Relative wound density was calculated after 36 h of cell stimulation.

### 2.6. Western Blotting Analysis

To investigate the activation of specific signaling proteins or PARP cleavage in cancer cells, MCF-7 cells were seeded in DMEM with 10% FBS and antibiotics on 6-well plates at a density of 2 × 10^5^ cells/well. After appropriate experimental treatment, proteins were extracted from the cells using a lysis buffer (containing 8% SDS, 2% β-ME), followed by sonication and heating. Proteins were then separated by SDS-PAGE electrophoresis and transferred onto a PVDF membrane using electroblotting. The membrane was probed with a primary antibody that specifically recognizes the protein of interest, followed by washing to remove the unbound primary antibody (listed in Section 2.1). To detect the protein–antibody complex, the HRP-conjugated secondary antibody was incubated with the membrane. Finally, the membrane was washed, incubated with HRP-substrate, and the proteins were visualized using ChemiDoc (BioRad, Hercules, CA, USA).

### 2.7. Cell Signaling Analysis

To evaluate the effects of growth factors on downstream signaling, serum-starved (24 h in DMEM with antibiotics) MCF-7 cells were treated with 10 ng/mL of FGF1 or EGF for specified times (0, 5, 15, 30, 120 min, and 6 h) in the presence or absence of the indicated inhibitors (20 µM UO126 or/and 20 µM LY294002). Inhibitors were added 15 min prior to stimulation. After the treatment, the cells were lysed using sample buffer (8% SDS, 2% β-ME), followed by sonication, heating, SDS-PAGE and Western blotting.

### 2.8. Statistical Analysis

Statistical analysis was performed using GraphPad Prism 5 (GraphPad Software, San Diego, CA, USA) with a one-tailed t-test, where a *p*-value less than 0.05 was considered statistically significant.

## 3. Results

### 3.1. FGF1 Protects MCF-7 Cells from Taltobulin in an FGFR-Dependent Manner

We first treated MCF-7 cells with several anticancer drugs that disrupt tubulin polymerization (5 nM taltobulin, 20 nM paclitaxel and 10 nM vincristine) in the presence or absence of 10 ng/mL FGF1 and analyzed cell viability using alamarBlue Cell Viability Reagent. Given that MCF-7 cells moderately express both FGFR and EGFR [10,13], we also investigated the effect of EGF (10 ng/mL) on the cytotoxicity of these drugs. For both growth factors, a reducing effect on cytotoxicity induced by all drugs tested was observed (Figure 1A), but the protection provided by EGF was weaker, especially for taltobulin (by 13.2%) and vincristine (by 11.7%) than for FGF1. It was also weaker for paclitaxel (by 8.5%) but without statistical significance. A control experiment, in which MCF-7 cells were treated with drug vehicle (DMSO) alone in the presence or absence of growth factors is presented in Appendix A. It should be mentioned that in our previous study, EGF did not exhibit a protective effect against these drugs in other EGFR-positive cells, such as DMS114 and HCC15 [11].

We then tested whether the protective effect of FGF1 and EGF depends on the activity of their receptors. We used specific inhibitors of both receptors, PD173074 for FGFR and gefitinib for EGFR. When RTK activity was blocked, the anti-cytotoxic action of growth factors was completely eliminated (Figure 1B).

In the next step, we investigated whether the pro-survival effect of both growth factors in taltobulin-treated MCF-7 cells is due to their anti-apoptotic action. For this purpose, cells were kept with 5 nM taltobulin for 24 h in the presence or absence of FGF1 or EGF. After lysis, PARP cleavage was assessed using a specific antibody. The results show that both proteins inhibit PARP processing, indicating that they prevent taltobulin-induced apoptosis (Figure 1C).

We also analyzed the effects of FGF1 and EGF on the migratory properties of MCF-7 cells treated with taltobulin by monitoring scratch overgrowth using the IncuCyte^®^ Cell Migration and Invasion System after 36 h. Exogenous addition of FGF1, but not EGF, altered cell migration patterns and significantly abolished the inhibitory effect of the drug on wound healing (Figure 1D, Appendix A). In a control experiment, we showed that both proteins do not change the migration pattern of cells not treated with the drug (Appendix A). This suggests that the role of this protein is not only to prevent apoptosis of tumor cells but also to protect their invasiveness relevant to metastasis. The effect of FGF1 was abolished by the addition of the FGFRs inhibitor, PD173074 (Figure 1D).

### 3.2. FGF1 Acts through Both ERKs and AKT to Protect MCF-7 Cells from Taltobulin

To further investigate the mechanism underlying the protection of MCF-7 cells against taltobulin and to identify the differences in the action of FGF1 and EGF proteins, MCF-7 cells were treated with taltobulin, alone or in conjunction with 10 ng/mL of FGF1 or EGF in the presence of specific inhibitors of cell signaling kinases, 20 µM LY294002 (PI3K inhibitor, upstream of Akt) and 20 µM UO126 (MEK inhibitor, upstream of ERKs). Cell viability was assessed 48 h later using the alamarBlue Cell Viability Reagent. The protective effect of both FGF1 and EGF was not affected by the inhibition of the upstream AKT activator, PI3K (Figure 2).

While inhibition of ERKs activation had no effect on FGF1 activity, it did eliminate the protective effect of EGF (Figure 2). To inhibit the protective effect of FGF1, it was necessary to block the activation of both ERKs and AKT kinases (Figure 2). This result may explain the previously observed differences in the pro-survival activity of FGF1 and EGF proteins (weaker EGF effect).

### 3.3. The Kinetics of Activation of ERKs and AKT in MCF-7 cells by FGF1 Is Prolonged Compared to EGF

To understand the differences between FGF1 and EGF in protection against taltobulin, we studied how they activate cell signaling pathways. MCF-7 cells were stimulated with 10 ng/mL FGF1 or EGF in the presence or absence of specific inhibitors of signaling pathways for 6 h. After the indicated times, the cells were lysed and the lysates were analyzed by Western blotting with specific antibodies to determine the activation status of individual cell signaling pathways.

FGF1 and EGF exhibit divergent kinetics of activation of cell signaling, particularly with regard to the activation of ERKs and AKT (Figure 3A). Notably, activation of ERKs induced by FGF1 was sustained for at least 6 h, whereas activation by EGF was stronger but transient, disappearing after 2 h (Figure 3A). FGF1 activated AKT in a cyclic manner, with two distinct peaks at 5 min and 2 h, while EGF triggered AKT activation that peaked at 5 min and then gradually subsided over 2 h (Figure 3A). Activation of p70S6 kinase, a direct substrate of mTOR and a marker of its activity [14], was time-shifted between FGF1- and EGF-stimulated cells; for FGF1, maximum activation occurred after 2 h and for EGF after 30 min. Furthermore, the activation of p70S6K persisted for 6 h after FGF1 stimulation, whereas for EGF, the activity declined after 6 h. In the case of FGF1, sustained activation occurred between 15 min and 2 h, whereas in the case of EGF, activation peaked after 30 min and then diminished (Figure 3A).

In the presence of UO126, the duration of ERK activity was reduced, for FGF1 to 2 h and for EGF to 30 min (Figure 3B). However, there was no complete inhibition of ERKs activation, which is consistent with previous reports where FGF1 reduced the inhibitory effect of UO126 [15]. The pattern of AKT activation by FGF1 was also altered, with AKT remaining active throughout the 6 h period, and no significant changes were observed for AKT activation by EGF (Figure 3B). Differences were also found in the activation of p70S6 kinase. Notably, FGF1 stimulation resulted in prolonged p70S6K activity relative to EGF-stimulated cells (Figure 3B).

Interestingly, inhibition of PI3K by LY294002 not only prevented AKT activation, but also altered the activation pattern of ERK1/2. This may suggest that AKT is involved in prolonged ERKs activation, which is essential for protection against taltobulin, and may also explain why both ERKs and AKT must be inhibited simultaneously to abolish the protective effect of FGF1.

We next examined the effects of a combination of PI3K and ERK inhibitors on the activation of signaling pathways by FGF1. When PI3K and ERKs were inhibited together, we observed complete suppression of p70S6 kinase activity, together with inhibition of AKT activation (Figure 3D). In addition, ERK phosphorylation was significantly reduced upon stimulation by FGF1 (also compared to the experiment using the inhibitor UO126 alone) (Figure 3D), confirming the previously described phenomenon of cross-activation of ERKs by AKT kinase [16,17].

## 4. Discussion

Breast cancer, a complex disease involving genetic and epigenetic alterations that dysregulate cell signaling pathways, is the leading cause of cancer death in women [18]. A key role in this cancer is played by the PI3K/AKT/mTOR signaling pathway, which regulates cancer cell survival, invasion and apoptosis [14]. Inhibitors targeting this pathway, such as everolimus and temsirolimus (mTOR inhibitors), are currently being evaluated in clinical trials [14]. Another important signaling pathway in breast cancer development and progression is Raf/MEK/ERK signaling cascade, which stimulates cell proliferation and gene expression and prevents apoptosis [19]. This pathway interacts with the PI3K/AKT pathway and their effects on cell growth, survival and drug resistance may vary depending on cell lineage and the presence of functional p53 and PTEN proteins [19]. These pathways are key in the regulation of cancer stemness, angiogenesis, metastasis and the interaction between tumor and stromal cells. Various growth factors and their receptors, including EGFRs, FGFRs, VEGFRs, PDGFRs and IGFRs are involved in the activation of these cellular signaling pathways, and their roles in breast cancer and drug resistance have been widely described [5,20]. Although they are attractive targets in breast cancer, therapies targeting receptor tyrosine kinases (RTKs) face challenges such as structural mutations of the receptors, amplification of their genes and activation of alternative signaling pathways that affect their efficacy [20].

In the previous work, we have shown that FGF1 can protect lung cancer (DMS114) and osteosarcoma (U2OSR1) cells, which overexpress FGFR1, from drugs targeting tubulin polymerization [11]. However, we found that in DMS114 cells and other EGFR-positive lung cancer cells such as HCC15, EGF administration did not affect the cytotoxicity of taltobulin. Protection by FGF1 occurs exclusively through AKT activation, but using two distinct mechanisms. Here, we aimed to investigate the effect of FGF1 on the MCF-7 cell line, which is widely used in research as a model of the A-luminal type of breast cancer [10]. Here, we showed that FGF1 protects MCF-7 cells from taltobulin, similar to what was observed in DMS114 and U2OSR1 cells. The protective effect of FGF1 was stronger than that of EGF. For both growth factors, it was completely eliminated when specific inhibitors of their respective receptors were used, indicating that the protection observed is receptor-dependent. Furthermore, we found that both FGF1 and EGF prevented taltobulin-induced apoptosis in MCF-7 cells, but only FGF1 altered cell migration patterns and abolished the inhibitory effect of taltobulin on wound healing. This suggests that FGF1 not only prevents apoptosis, but also restores cell migration relevant to invasiveness and metastasis. Previous studies have shown that both FGFR1 and EGFR are involved in the regulation of EMT-related gene in the context of drug resistance induced by paclitaxel, doxorubicin and docetaxel in MCF-7 cells [21]. However, the fact that only the action of FGF1/FGFR is able to overcome the inhibitory effect of taltobulin on cell migration suggests a different mechanism of protective action of FGF1 and EGF and their receptors in the case of taltobulin.

We also investigated the signaling pathways involved in the protective effect of FGF1 and EGF. A previous study in other cell lines has shown that AKT is a key player in FGFs/FGFRs-dependent protection against drugs targeting tubulin polymerization [11,22]. In the MCF-7 line, the protective effect of FGF1 and EGF against taltobulin was not affected by inhibition of the upstream AKT activator, PI3K. Inhibition of ERKs activation had no effect on FGF1 activity, while it completely abolished the protective effect of EGF against the drug. Suppression of both ERKs and AKT kinase was required to block the protective effect of FGF1. These observations could be attributed to the prolonged activation of mTOR substrate, p70S6K, in the presence of FGF1 compared with EGF. During ERK inhibition by UO126, FGF1 continued to activate the substrate of mTOR, whereas EGF did not. On the other hand, PI3K inhibition abrogates p70S6K activation by both FGF1 and EGF. Interestingly, we found that when PI3K is blocked, the pattern of ERKs activation by FGF1 and EGF is similar, which may suggest a role for the PI3K/AKT/mTOR pathway in ERKs reactivation independent of MEK inhibition by UO126. This demonstrates that FGF1 has a different mechanism of action in protecting MCF-7 cells against taltobulin compared to DMS114 or U2OSR1 cells and that the mechanisms of action of FGF1 and EGF differ (Figure 4).

In summary, FGF1, EGF and their receptors are important players in breast cancer development. The stronger protective effect of FGF1 than EGF against taltobulin and its effect on MCF-7 cell migration suggests its particular relevance in tumor progression and drug resistance. These insights highlight the importance of FGF1 in overcoming taltobulin-induced effects and suggest therapeutic implications. By understanding the different t actions of FGF1 and EGF on individual cancer cell types, targeted approaches should become possible to develop new effective treatment approaches while preventing the acquisition of drug resistance and cancer progression.

A combination of strategies based on metabolic checkpoint blockade, growth factor inhibition (e.g., via ligand traps [23,24]), and multilevel suppression of mTOR [25,26] appears to have great potential for cancer treatment by simultaneously reprogramming cancer cell metabolism, enhancing the immune response and inhibiting cancer cell proliferation and survival.

Understanding the interplay between FGF1 signaling pathways may pave the way for the development of more effective targeted therapies for breast cancer patients. Undoubtedly, however, further research is required to fully elucidate the role of FGF1 in breast cancer biology and its potential as a therapeutic target.

## 5. Conclusions

Our study advances the understanding of the complex interactions between growth factors, signaling pathways and drug resistance mechanisms in breast cancer. In particular, we have demonstrated for the first time the protective role of FGF1 in MCF-7 breast cancer cells against cytotoxicity induced by taltobulin, which is mediated by the activation of FGFRs and subsequent ERKs and AKT activation. These findings suggest that targeting FGF receptor ligands (e.g., in the form of ligand traps) or the signaling pathways they activate (via specific inhibitors) should be considered when designing new therapies for breast cancer in the context of counteracting chemoresistance.

## Figures and Tables

**Figure 1 biomedicines-11-01856-f001:**
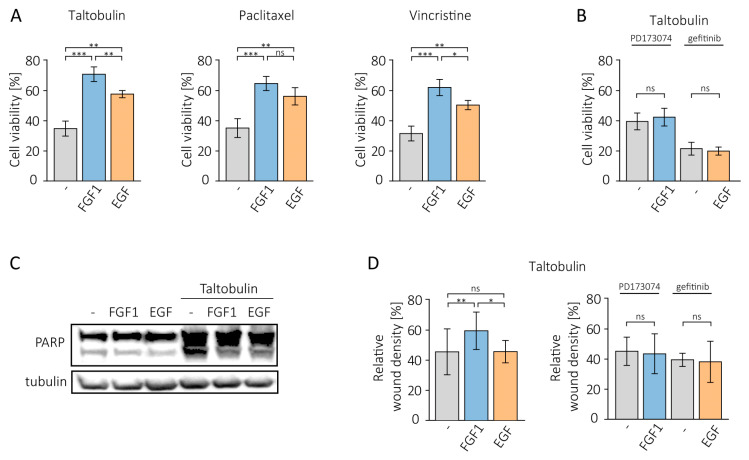
Protective effect of FGF1 and EGF in MCF-7 cells treated with cytotoxic drugs. (**A**) The effect of FGF1 and EGF stimulation (10 ng/mL) on drug-induced cytotoxicity was investigated in MCF-7 cells treated with 5 nM TLT, 20 nM PTX or 10 nM VCR for 48 h. Cell viability was assessed using the alamarBlue assay. (**B**) The effect of growth factor receptor inhibition on FGF1 and EGF activity against taltobulin cytotoxicity was tested using 100 nM PD173074 and 10 µM gefitinib. (**C**) The anti-apoptotic effect of FGF1 and EGF against taltobulin-induced apoptosis in MCF-7 cells was assessed by evaluating PARP cleavage by Western blotting. Anti-PARP antibodies were used to detect PARP cleavage 24 h after treatment with 5 nM TLT in the presence or absence of 10 ng/mL FGF1 or EGF. (**D**) The effect of FGF1 and EGF on MCF-7 cell migration in the presence of 5 nM taltobulin was examined after 36 h using IncuCyte^®^ Cell Migration and Invasion System. Normalization of relative wound density was based on both the density of cells in the wound area and the width of the wound itself and is expressed as a percentage of the wound area that was filled by migrating cells over time. Data are presented as mean values ± standard deviation (SD) from three independent experiments. Statistical significance: * *p* < 0.05, ** *p* < 0.01, *** *p* < 0.001, no significant differences indicated as ‘ns’.

**Figure 2 biomedicines-11-01856-f002:**
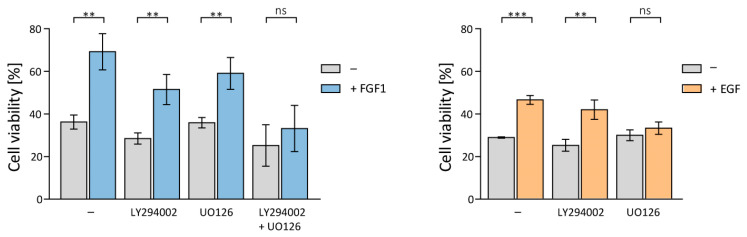
Impact of inhibition of PI3K/AKT/mTOR and MAPKs signaling pathways on the protective effect of FGF1 and EGF against taltobulin. The viability of MCF-7 cells treated with 5 nM TLT and various chemical inhibitors targeting key signaling pathways (20 µM LY294002 for PI3K, 20 µM UO126 for MEK1/2, and a mixture of 20 µM LY294002 and 20 µM UO126 for both PI3K and MEK) was measured after 48 h in the presence or absence of 10 ng/mL FGF1 or EGF using the alamarBlue assay. Data were normalized to untreated cells and are presented as mean values ± standard deviation (SD) from three independent experiments. Statistical significance: ** *p* < 0.01, *** *p* < 0.001, no significant differences indicated as ‘ns’.

**Figure 3 biomedicines-11-01856-f003:**
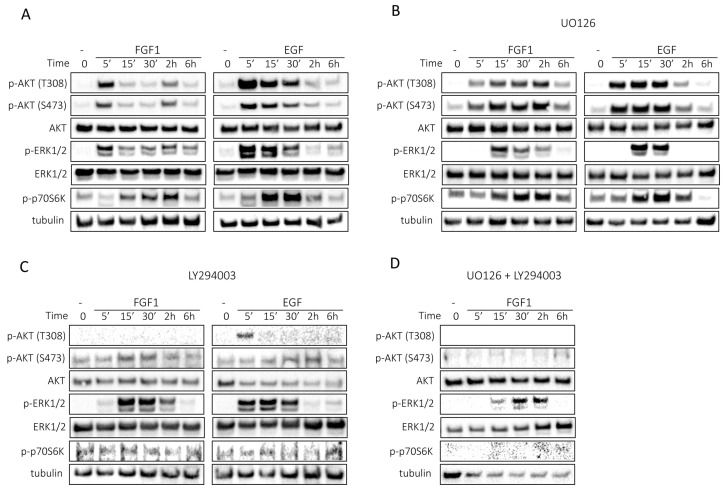
Kinetics of FGF1- or EGF-stimulated cell signaling in MCF-7 cells. Serum-starved MCF-7 cells were treated with 10 ng/mL FGF1 or EGF in the absence (**A**) or presence of MEK inhibitor (20 µM UO126) (**B**) or presence of PI3K inhibitor (20 µM LY294002) (**C**) or a mixture of MEK inhibitor (20 µM UO126) and PI3K inhibitor (20 µM LY294002) (**D**) for different times: 0, 5 min, 15 min, 30 min, 2 h and 6 h. Subsequently, cell lysates were subjected to Western blotting with specific antibodies to assess the activation of cellular signaling pathways, including AKT/mTOR and ERKs.

**Figure 4 biomedicines-11-01856-f004:**
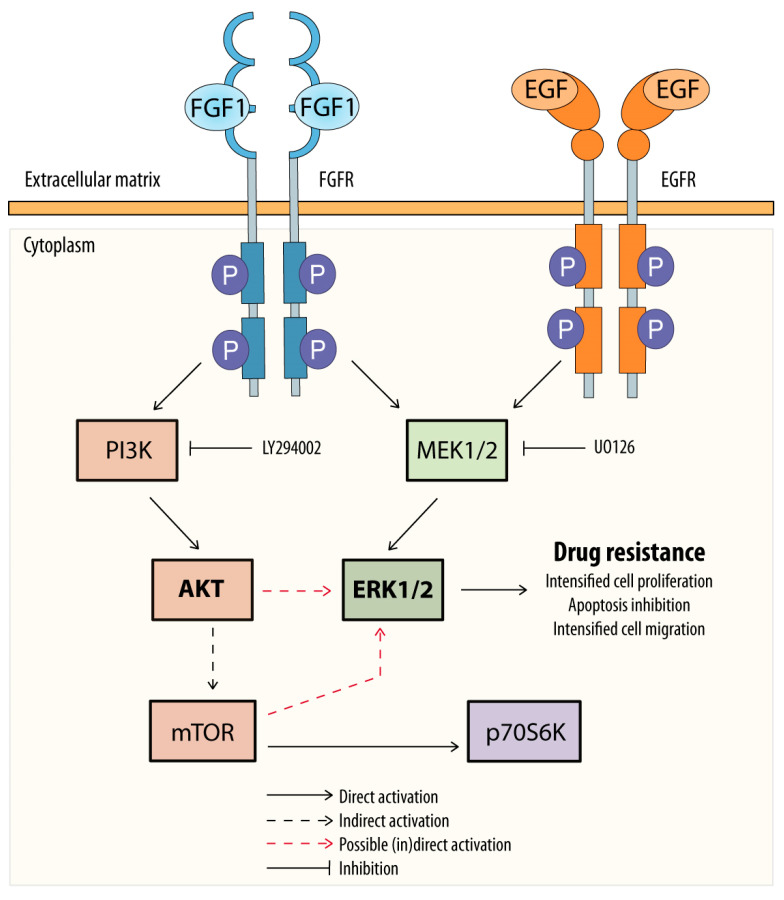
Potential mechanism of FGF1 and EGF action in protecting MCF-7 cells against taltobulin. Both FGF1 and EGF, upon activation of their receptors (FGFR or EGFR), protect MCF-7 cells via the MEKs/ERKs pathway, and MEK1/2 inhibition by UO126 blocks this action. However, in the case of FGF1, alternatively, protection of MCF-7 cells against taltobulin can occur through activation of the AKT/mTOR pathway, which then reactivates ERKs.

## Data Availability

Data available within the article and its Appendix A.

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
