# Peer review of "FGF1 Protects MCF-7 Cells against Taltobulin through Both the MEKs/ERKs and PI3K/AKT Signaling Pathway"

_biomedicines, 2023, doi:10.3390/biomedicines11071856_

Round 1

Reviewer 1 Report

The paper titled "FGF1 protects MCF-7 cells against taltobulin through both the 2 MEKs/ERKs and PI3K/AKT signaling pathway" aims to investigate the impact of FGF1 compared to EGF on the survival of the MCF-7 cell line, which is widely used in breast cancer research. The study focuses on analyzing the effects of FGF1 and EGF on cell viability, apoptosis progression, and cell migration in the presence of taltobulin, a commonly used cytotoxin that inhibits tubulin polymerization.

In summary, to improve the paper, provide clear descriptions of the experimental methods, discuss the potential underlying mechanisms, compare the effects of FGF1 and EGF, and address the clinical implications of the findings. By addressing these aspects, the paper will become more informative, comprehensive, and impactful in the field of breast cancer research.

Reviewer 2 Report

In the paper named “FGF1 protects MCF-7 cells against taltobulin through both the 2 MEKs/ERKs and PI3K/AKT signaling pathway”Author demonstrates a protective role for FGF1 in MCF-7 breast cancer cells against taltobulin induced cytotoxicity, mediated by activation of its receptors and compares its activity to EGF, another growth factor involved in breast cancer development and progression. Therefore this study advances in the understanding of the complex interactions between growth factors, signaling pathways and drug resistance mechanisms in breast cancer.

Only minor points are required

1)In material and methods section information about the inhibitors used in the paper are missing, none information about the inhibitors concentration or about how long before the treatment is given this inhibitors...

2) In material and methods section information about the antibodies used in western blot are missing.

3) It will be interesting known why authors used the 5 nM taltobulin, 20 nM paclitaxel, and 10 nM vincristine doses

4)In line 136 EGF dose is missing

5) Although authors state that EGF protection (figure 1A) was weaker especially in vincristine and taltobulin it seems that in paclitaxel the protective effect it is weak too, please can author include the % of protection?

6) In figure 1C author say that both FGF1 and EGF shown a protective effect against PARP cleavage, however in this figure a high amount of parp can be seen after Taltobulin treatment. Can this effect due to the cleavage inhibition?

7) In figure 1D an image of the migration effect may be added

8) Why in point 3.3 authors used a 10 times higher dose of EGF1 and EGF?

9) In figure 3 author only shown the western of the phosphorilated proteins however the amount of total protein is needed in order to known if the total protein are not altered by the treatment.

Minor editing of English language required

Reviewer 3 Report

This article tries to evaluate the protective roles of FGF1 and EGF in the cytotoxicity induced by various anti-cancer drugs, including taltobulin, a tubulin polymerization inhibitor. The authors found that FGF1 exerts its protective effects through the activation of both ERKs and AKT, whereas EGF acts only through ERKs. Overall, the experimental designs were too simplified to achieve what was proposed and support the conclusion made in the title.

1. Materials and Methods were oversimplified, lacking many essential details as to how experiments were exactly performed. Were cells seeded in dishes in serum-free culture media or in media containing FBS?

2. Isn't it true or naturally expected that any growth factors, just like FGF1 and EGF, will exert protect role against anti-cancer drugs by promoting cell proliferation and growth? What were the big deals for the results in this article?  I don't any specificity particularly for the role of FGF1, unless the authors were able to exhibit that other growth factors fail to protect cancer cells from the cytotoxicity of anti-cancer drugs.

3. In Fig. 1 and 2, to make solid conclusions, all the experiments need proper controls where the cells for -, +FGF1, or +EGF treatments should be treated with the vehicle in the absence of Taltobulin, paclitaxel, or Vincristine for the individual normalizations of cell viability and wound density.

4. In Fig. 3, all the IB images should include cells treated with taltobulin. Moreover, the control in which cells are treated with LY294002 alone is missing. These results, therefore, do not support the idea that the authors wanted to deliver in this study.

Reviewer 4 Report

Dear Authors, the manuscript is well organized and the results is sufficient for brief report. For my opinion the manuscript is eligible for pubblicatio. 

Reviewer 5 Report

Article by Dr. Zakrzewska and group elaborating on the role of  FGFs and their receptors in promoting drug resistance in  FGFR-driven tumors. This is a well-experimental designed research article that follows through the hypothesis of the work. However, a few things must be addressed before it is ready for acceptance. They are as follows:

1. Authors need to use any other cell line to prove the findings from MCF7 

2. It will be better if authors add a model in the last figure- summarizing the take-home message from this paper

3. Also, add a few lines in the discussion part about the mTOR-regulated metabolic checkpoints, which have been discussed in PMID: 26682255 and PMID: 30131808 as a future aspect of this manuscript. 

Round 2

Reviewer 3 Report

My main concern remains as illustrated in the following 3 comments:

1. Comparing Fig. 1A and supplementary Fig. 1A, I find that Taltobulin still triggered a significant decrease of cell viability even in the presence of FGF1 (~from 130% to 70%, a 60% decrease) or EGF (~from 120% to 60%, a 60% decrease) that was similar to the reduction caused by Taltobulin in the absence of both growth factors (~100% to 40%, a 60% decrease), meaning that FGF1/EGF treatments did not really protect tumor cells against the Taltobulin assault. Thus, these results do not sufficiently support the conclusion made in the title.

2. Without comparing the inhibitory effects of AKT and MEK inhibitors on cell viability in the absence of Taltobulin in Fig. 2 (not even provided in the supplementary results), how did you exclude that the inhibitor effects only inhibited the cell viability increased by FGF1/EGF that had nothing to do with taltobulin treatment? These results again do not support the title.

3. I agree that the idea behind the experiments in Fig. 3 was to verify differences in the activation of FGFR- and EGFR-dependent signaling pathways shortly after growth factor stimulation, which the authors believe is crucial for further cell protection. However, "believing" is far from "proving" and can only serve as a hypothesis. The authors did not provide any supporting evidence AT ALL to make me "believe" that FGF1 uses both pathways to protect tumor cells against taltobulin. The emphasis made by the authors that they do not observe differences independently of the presence of taltobulin in the analyzed time intervals EXACTLY STRENGTHENS AGAIN (echoing my comments #1 and 2) that both MEKs/ERKs and PI3K/AKT pathways are only possibly involved in the enhancement of tumor cell viability upon FGF1/EGF treatments, which entirely has nothing to do with any protective effects against taltobulin assault that is, after all, the focus of the title.

Reviewer 5 Report

All concerns have been addressed, ready for acceptance.